# 30-Minute Highly Multiplexed VaxArray Immunoassay for Pneumococcal Vaccine Antigen Characterization

**DOI:** 10.3390/vaccines10111964

**Published:** 2022-11-19

**Authors:** Tianjing Hu, David F. Miller, Amber W. Taylor, Christine Riley, Caitlin McCormick, Keely N. Thomas, Rachel Y. Gao, Kathy L. Rowlen, Emilia B. Byrne, Pardeep Kumar, Soo Kyung Kim, Erica D. Dawson

**Affiliations:** 1InDevR Inc., 2100 Central Ave., Suite 106, Boulder, CO 80301, USA; 2Pfizer Inc., 875 Chesterfield Pkwy W, Chesterfield, MO 63017, USA; 3R&D Center, EuBiologics Co., Ltd., 125, Wonmudong-gil, Dongsan-myeon, Chuncheon-si 24410, Gangwon-do, Republic of Korea

**Keywords:** pneumococcal polysaccharide vaccine, pneumococcal conjugate vaccine, CRM197, identity testing, antigen quantification, vaccine characterization, multiplexed immunoassay, microarray, VaxArray

## Abstract

Pneumonia accounts for over 20% of deaths worldwide in children aged 1 to 5 years, disproportionately affecting lower- and middle-income countries. Effective, highly multivalent pneumococcal vaccines are available to decrease disease burden, with numerous new vaccines currently under development to serve a variety of worldwide markets. However, pneumococcal conjugate vaccines are among the hardest biologics to manufacture and characterize due to their complexity and heterogeneity. Current characterization methods are often inherently singleplex, requiring separate tests for each serotype present. In addition, identity and quantity are often determined with separate methods. We developed the VaxArray pneumococcal assay for applications in identity, quantity, and stability testing of pneumococcal polysaccharide and pneumococcal conjugate vaccines. The VaxArray pneumococcal assay has a time to result of less than 30 min and is an off-the-shelf multiplexed, microarray-based immunoassay kit that can identify and simultaneously quantify 23 pneumococcal polysaccharide serotypes common to many on-market and in-development vaccines. Here, we highlight the potential of the assay for identity testing by showing high reactivity and serotype specificity to a wide variety of native polysaccharides, CRM197-conjugated polysaccharides, and drug product. The assay also has vaccine-relevant lower limits of quantification in the low-to-mid ng/mL range and can be used for accurate quantification even in adjuvanted vaccines. Excellent correlation to the anthrone assay is demonstrated, with VaxArray resulting in significantly improved precision over this antiquated chemical method.

## 1. Introduction

Over 740,000 pneumonia-related deaths occurred worldwide in 2019 in children under the age of 5, accounting for 22% of worldwide deaths in children aged 1 to 5 years [1], with disproportionately higher incidence in lower- and middle-income countries [2,3,4]. Streptococcus pneumoniae is the most common causative agent of pneumonia [1]. While this common bacterium is part of the normal flora of the human nasopharynx [5], it causes a variety of mild diseases, including otitis media and sinusitis with significant associated morbidity [4,5], and invasive diseases, including bacterial meningitis, sepsis, and peritonitis [6,7]. There are over 100 unique serotypes of *S. pneumoniae*, with ~23 serotypes causing most cases of invasive disease [7].

The importance of pneumococcal vaccines, particularly pneumococcal conjugate vaccines, cannot be overstated in terms of the impact of reducing disease burden. Pneumococcal polysaccharide vaccines (PPVs), with a 23-valent formulation first introduced in 1983, elicit poor immune responses in young children due to the absence of an associated T-cell response but are still effectively used in adults and the elderly [8,9]. The introduction of pneumococcal conjugate vaccines (PCVs) in 2000 represented an important turning point in the fight against pneumococcal disease in children [10].

PCVs involve the chemical coupling of the main virulence factor of *S. pneumoniae*, the purified pneumococcal capsular polysaccharide, to a so-called carrier protein such as the mutant diphtheria toxoid cross-reactive material 197 (CRM197) that produces a robust T-cell-dependent antibody response [8,9]. These combined conjugates elicit robust immunity in children. Since the introduction of PCV7 (Prevnar^®^) in 2000 and subsequent WHO recommendations to add PCV to national vaccination schedules [3,11], a variety of PCVs with increased valency have been introduced to the market to offer vaccines at reduced prices for low-income countries [3,12]. Importantly, the 2019 WHO prequalification of Serum Institute of India’s Pneumosil vaccine represents another significant achievement in increasing access and reducing price of PCVs for lower- and middle-income countries [13,14]. Because serotype prevalence varies by region [4,8,15], serotype coverage in vaccines on-market and in-development can vary based on the intended population [16].

Worldwide, a number of PCVs are currently in development and are in various stages of clinical studies [4]. It is acknowledged that conjugate vaccines, including PCVs, are among the most complicated biologics to manufacture in part due to complex glycoconjugate attachment chemistry, other structural complexities and heterogeneities [17], and high valency. For example, a review of Pfizer’s 2015 annual report indicates that producing Prevnar 13TM requires 580 manufacturing steps and 678 associated quality tests, with an overall manufacturing time of 2.5 years [18]. Given these complexities, a significant amount of the manufacturing effort is spent on performing quality testing on the various raw materials, intermediates throughout multiple stages of manufacturing, drug substance, and final drug product.

Various analytical methods are currently utilized for identity testing, quantification, and stability testing of PCVs. For identity testing, nuclear magnetic resonance (NMR) is commonly used on bulk polysaccharide but not on conjugates due to the presence of the carrier protein [17,19,20]. NMR may be commonplace and available in high-income countries and with established manufacturers, but the associated instrumentation or outsourcing this testing may be prohibitively expensive in lower- and middle-income countries. Alternatively, immunoassay-based methods such as dot blot, ELISA, and rate nephelometry are utilized for identity testing of the conjugate material [17,19,20]. Most of these methods are time-consuming, labor-intensive, and run in a singleplex manner, requiring separate tests for each serotype. Because PCVs now include as many as 20 serotypes or more, utilizing the singleplexed assays for qualifying the highly multivalent drug products takes considerable time and effort.

Quantification of the native polysaccharides is typically performed using a generic chemical assay, such as the anthrone assay with a colorimetric readout [17,19,20], but these assays are hazardous (employing concentrated sulfuric acid to elicit the colorimetric readout) and, more importantly, non-specific; therefore, they cannot be employed on multivalent material. High-performance anion-exchange chromatography with pulsed amperometric detection (HPAEC-PAD) has also been employed as a quantification method with high sensitivity; however, samples require hydrolysis and carrier protein release prior to analysis, and the equipment is still relatively expensive [21,22,23]. Alternatively, the immunoassay-based methods mentioned previously may be used with suitable standards for quantification but come with the aforementioned drawbacks in addition to reagent development burden and its associated increased animal use. While multiplexed immunoassay-based methods such as those involving bead-based technology have been reported for direct antigen detection in urine for diagnostic purposes and can be developed for vaccine antigen characterization [24,25,26], the lack of availability of off-the-shelf, validated kits places much of the burden of assay development and optimization on individual manufacturers.

In this work, we present analytical performance metrics of a new 23-valent VaxArray pneumococcal assay available as a pre-validated, off-the-shelf kit for identity, quantity, and stability testing of pneumococcal polysaccharide and pneumococcal conjugate vaccines from monovalent drug substance through highly multivalent drug product. The VaxArray pneumococcal assay can be executed in less than 30 min from initial sample prep through final data analysis and offers “off-the-shelf” availability, an advantage that can improve lab-to-lab and site-to-site consistency over currently utilized methodologies, including in-house-developed immunoassay-based methods. Here, we highlight performance metrics such as specificity, analytical sensitivity, accuracy, and precision, including in the presence of adjuvant, on a wide variety of native polysaccharides, conjugates, and final on-market vaccines to demonstrate the capabilities of the assay.

## 2. Materials and Methods

### 2.1. Antibodies

Monoclonal capture antibodies raised against serotype-specific pneumococcal polysaccharides (PS) were obtained from sources, including PATH (under an MTA), NIBSC (Hertfordshire, UK), Immune Diagnostics, Inc (Asheville, NC, USA), and AbMax Biotechnology Co., Ltd. (Beijing, China). Serotype-specific polyclonal anti-PS antisera used as detection labels were purchased from Cedarlane Laboratories (Burlington, ON, Canada). Anti-Diphtheria Toxoid CRM197 antibody was obtained from Alpha Diagnostic Intl., Inc, (San Antonio, TX, USA). Alexa Fluor^TM^ Plus 555 conjugated anti-rabbit secondary antibody was purchased from Thermo Fisher Scientific (Waltham, MA, USA).

### 2.2. VaxArray Pneumococcal Assay

The VaxArray pneumococcal assay is similar to previously described VaxArray assays [27,28,29], with the slide layout, microarray layout, and detection principle depicted in Figure 1a–c, respectively. Pneumococcal capsular polysaccharide-specific monoclonal antibodies are printed on the microarray (Figure 1b) and used to capture polysaccharides (native or conjugate materials), which are then detected by a primary–secondary label pair. Each kit (VXPN-9000, InDevR Inc., Boulder, CO, USA) contains 2 microarray slides, each with 16 replicate arrays (Figure 1a,b), an optimized blocking buffer (PBB), and two Wash Buffer concentrates (WB1 and WB2). VaxArray slides were first equilibrated at 25 °C for 30 min and placed in the ArrayMax^TM^ mixer (VX-6212, InDevR Inc.). Individual arrays were pre-washed with 45 µL 1× WB1 on ArrayMax at 700 rpm for 1 min at 25 °C, after which samples (diluted in PBB as needed) were applied to designated arrays. Slides were incubated statically for 5 min at 25 °C. All subsequent wash and label incubation steps used the ArrayMax at 700 rpm. Arrays were then washed with 1× WB1 for 1 min, incubated with Primary Detection Label (selected from VXPN-7671 to VXPN-7695, InDevR Inc.) for 5 min, washed with 1x WB1 for 1 min, incubated with secondary detection label (VXPN-7670, InDevR Inc.) for 5 min, and subjected to sequential washes with 1× WB1, 1× WB2, 70% ethanol, and water. Slides were dried and imaged using the VaxArray Imaging System (VX-6000, InDevR Inc.), and images were processed using the VaxArray Analysis Software (v2.2, InDevR Inc). When appropriate, sample concentrations were calculated against a standard curve from the same experiment.

### 2.3. Detection Label Optimization

Primary and secondary detection label concentrations were optimized. To ensure adequate reactivity, specificity, limit of detection, and dynamic range, several dilutions of each serotype-specific anti-PS antiserum was tested on serial diluted monovalent samples in combination with different incubation methods and times. Based on the signal response on target capture antibody and off-target capture antibodies, initial dilutions of anti-PS antisera were chosen to generate a polysaccharide label mixture. This mixture was tested on both multivalent and monovalent native PS and conjugates, and dilutions were further fine-tuned for each serotype in the final 23-Mix Primary Detection Label (VXPN-7671) to enable simultaneous 23-valent PS detection.

### 2.4. Pneumococcal Samples and Vaccines

Purified individual native polysaccharides (native PS), CRM197 conjugated polysaccharides (conjugate), and mock drug products were kindly provided by Pfizer (St. Louis, MO, USA) and EuBiologics Co., Ltd. (Seoul, Republic of Korea), and native PS (deposited by Pfizer) were also purchased from ATCC (Manassas, VA, USA). Vaccines including Prevnar 20^TM^ (Pfizer) and PNEUMOVAX^®^ 23 (Merck) were purchased from Global Sourcing Initiative (Miami, FL, USA).

### 2.5. Reactivity/Specificity

Serotype specificity of the 23 capture antibodies was verified with monovalent native PS and conjugate analyzed at 2 μg/mL for each serotype to ensure any low-level cross-reactivity between serotypes would be observed. A “label-only” blank (primary and secondary detection label pair with no added antigen) was also analyzed to assess any direct binding of the detection labels to the captures. To assess reactivity and specificity, signal-to-blank (S/Bl) ratios were calculated, with S/Bl > 3 considered reactive (targeted capture) or cross-reactive (off-target captures) for a given monovalent sample.

### 2.6. Analytical Sensitivity and Dynamic Range

To estimate lower and upper limits of quantification (LLOQ and ULOQ), multivalent samples were generated by combining individual native PS or conjugates of interest at desired stock concentrations. From the stock, three replicate 16-point dilution series were prepared and processed as described above. A moving four-point linear fit was applied to each dataset, with slope and R^2^ for each regression calculated. After mapping the concentration range over which linear fits produced R^2^ > 0.9, LLOQ was estimated by determining the concentration at which the signal generated was equal to the average blank signal + 5σ (where σ = standard deviation of the background signal). ULOQ was approximated by determining the concentration at which the signal was 90% of the maximum observed signal. The dynamic range of the assay was expressed as ULOQ/LLOQ.

### 2.7. Citrate Buffer Desorption of Vaccines

Vaccines were mixed with sodium citrate tribasic dihydrate solution (C8532, Sigma-Aldrich, St Louis, MO, USA), pH = 9, at a final concentration at 200 mM and placed in a 25 °C incubator for 16 h. The treated samples were then centrifuged at 1500× *g* for 5 min at room temperature, and the supernatant was carefully collected for analysis.

### 2.8. Assay Precision and Accuracy

For native PS, three users analyzed eight replicates of a contrived 23-valent PS (ATCC, deposited by Pfizer) over each of 3 days (3 users × 8 replicates × 3 days = 72 replicates) using the Pneumo 23-Mix Primary Detection Label (VXPN-7671, InDevR Inc.) and the Pneumo Secondary Detection Label (VXPN-7670, InDevR Inc.). On each testing day, an aliquot of the 23-valent sample was used to create a serial dilution from which a standard curve was generated. In addition, a check standard to be run in replicate was created from the same aliquot by diluting the sample 4x in PBB prior to analysis. Individual and user-aggregated precision were quantified and expressed as average %RSD of replicate measurements over all 3 days. Accuracy was calculated as % of expected concentration (measured value divided by expected value, expressed as a percentage) and quantified for each user and in aggregate.

For final vaccine accuracy and precision, single-user studies using PNEUMOVAX^®^ 23 and Prevnar 20^TM^ were conducted over three days, generating n = 24 replicates for each vaccine. Prevnar 20^TM^ was sodium citrate desorbed as described above prior to analysis, with replicate samples under evaluation and the standard curve prepared from same desorbed material. On each day, eight replicates of vaccine material were prepared at 0.2 μg/mL in each serotype present based on the product insert (0.4 μg/mL for S6B in Prevnar 20^TM^) and measured against matching calibration curve.

### 2.9. Comparison to Anthrone Assay

Anthrone reagent was prepared fresh by dissolving 1 g anthrone (Sigma-Aldrich, St. Louis, MO, USA) in 500 mL concentrated sulfuric acid. A 12-point dilution series of the PS of interest was prepared, and the assay was performed in 96-well plate format [30]. Briefly, a 100 µL sample was mixed with 200 µL anthrone reagent, with each dilution analyzed in triplicate. The plate was incubated at 95 °C for 20 min and scanned at 630 nm on a FLUOstar OPTIMA microplate reader (BMG LABTECH, Ortenberg, Germany). Because VaxArray has improved sensitivity over the anthrone assay, serial dilutions prepared for anthrone assay analysis were diluted an additional 100× or 200× prior to VaxArray analysis. The samples were assessed by VaxArray in triplicate, as described above, using individual serotype-specific Primary Detection Labels. Correlations were generated by plotting the anthrone assay blank-corrected against the VaxArray signal response (median RFU minus background) for each sample.

### 2.10. Statistical Analysis

Statistical significance (*p* < 0.05) was determined using unpaired *t*-test (GraphPad Prism 9.3.1). Percent relative standard deviation for replicate measurements (%RSD) was calculated as the standard deviation divided by the mean, expressed as a percentage. Accuracy or percent recovery was calculated as the measured value divided by the expected value, expressed as a percentage.

## 3. Results and Discussion

### 3.1. Antibody Screening and Detection Label Optimization

Antibodies against pneumococcal polysaccharides serotypes 1, 2, 3, 4, 5, 6A, 6B, 7F, 8, 9N, 9V, 10A, 12F, 14, 15B, 17F, 18C, 19A, 19F, 20, 22F, 23F and 33F were obtained and printed in triplicate on a “screening” microarray to evaluate reactivity, specificity, and dynamic range. Serotypes were chosen for inclusion that are most common to a wide variety of pneumococcal conjugate vaccines on market and in development. While serotype 11A was targeted for inclusion, poor antibody performance and lack of high-quality alternatives available at the time of development precluded its inclusion. The final array layout with a single capture antibody for each serotype, each printed in three replicate spots, is shown in Figure 1b. Detection labels were developed and optimized for labeling either the polysaccharide or the CRM197 (Figure 1c), as described in the Materials and Methods section. The fluorescently conjugated secondary detection label utilized was the same for both PS and CRM197 labeling strategies.

### 3.2. Reactivity and Specificity

Figure 2 shows representative fluorescence images of monovalent PS and contrived 23-valent PS (ATCC, deposited by Pfizer) labeled using the 23-Mix Primary Detection Label. These images qualitatively indicate that the serotype-specific antibodies generate specific signal responses and that the off-target antibodies do not produce a signal significantly above the label-only/no-antigen blank. As shown in Table 1, at 2 μg/mL, all tested monovalent native PS generated signal-to-blank ratios (S/Bl) on the target capture antibody ≥ 3, indicating reactivity, with 9 of 23 serotypes producing S/Bl ratios > 13. In contrast, all but one off-target capture antibodies produced S/Bl ≤ 1.6, which varied slightly between serotypes, indicating no appreciable non-specific signal, with one exception in that the S14 capture antibody cross-reacts with S15B native PS, resulting in S/Bl = 3.2. Of note, other native PS and conjugate materials tested did not exhibit this same cross-reactivity of S15B on the S14 capture (data not shown). All 23 individual native PS were also mixed at 2 μg/mL and tested as a 23-valent mixture (lower right image in Figure 2), demonstrating the potential of the assay for simultaneous analysis of up to 23 serotypes in a multivalent mixture.

In Figure 3a, we highlight eight-point response curves with concentration starting at 0.8 μg/mL as tested for serotypes 1, 5, 10A, and 19F; analyzed monovalently using the 23-mix PS labeling strategy; and presented as signal-to-blank values. Each of the four serotypes shown demonstrates good linearity with all R^2^ ≥ 0.92. We also assessed signal response for 23 serotypes of native PS (ATCC, deposited by Pfizer) analyzed in both monovalent and multivalent form, as shown in Figure 3b. Each PS serotype represented on the microarray was tested at 0.8 μg/mL both mono- and 23-valently with the 23-Mix Primary Detection Label. Monovalent and multivalent signal-to-blank (S/Bl) ratios for each serotype were compared via unpaired *t*-tests. Statistically equivalent signals were observed between the mono- and multivalent samples for all serotypes. To assess response similarity over a broader concentration range, all 23 serotypes were tested at three different concentrations. Similar responses were observed at all tested concentrations. Data for four representative serotypes are shown in Figure 3c, indicating comparable responses over the three tested concentrations for both monovalent and multivalent analyses. Given this similarity, the 23-valent mixture was employed in all subsequent experiments. While these data show no interference in a multivalent sample due to the presence of the other serotypes, for ideal quantification, we still recommend using a matched standard (in terms of valency and matrix) whenever possible.

Reactivity and specificity were also assessed for monovalent conjugates (13 Pfizer-provided and 14 EuBiologics-provided) using both the 23-Mix PS (Appendix A) and CRM197 labeling strategies (Appendix A). All targeted serotypes, with two exceptions, generated S/B ratios ≥ 3 on the corresponding target capture antibody with both labeling strategies, while all off-target capture antibodies S/Bl ratios were ≤2.2 at 2 μg/mL. The 23-Mix PS label was not reactive to the Pfizer-provided S18C conjugate. Given the anti-CRM197 label data (Appendix A) that indicate successful detection of this same S18C conjugate when labeled via the CRM197 and evidence that the same monoclonal capture antibody can be successfully used as the detection label (data not shown), the material is clearly being captured on the array but not effectively labeled via the polysaccharide with the specific antisera label being utilized. We also note that a different lot of S18C native PS, as shown in Table 1, is recognized and that the reason for the 23-Mix PS label not detecting this specific material is unknown. In addition, structures of conjugates from different sources may be distinct, and an optimized antibody raised to the specific conjugate maybe needed. Although not observed with native S19A polysaccharide or one S19A conjugate (EuBiologics-provided), the S23F capture antibody elicited cross-reactivity to the other S19A conjugate (Pfizer-provided) independent of detection strategy. It may be that distinct assay conditions or an optimized antibody raised to the specific material in question are required for suitable detection of these materials.

Target specificity is a critical attribute of any assay to be utilized as an identity test, and with a few exceptions for specific materials, as noted above, these data demonstrate overall good reactivity and serotype specificity, indicating that the VaxArray pneumococcal assay is suitable as a rapid identity verification test for both monovalent and multivalent samples. One potential application of this technology is as a quality assurance tool when material is shipped and received to various sites, including drug products shipped to regulatory authorities.

### 3.3. Limits of Quantification and Dynamic Range

The ULOQ and LLOQ were estimated for the same 23-valent native PS materials described above (ATCC, deposited by Pfizer), with dynamic range defined as ULOQ/LLOQ. As shown in Table 2, most serotypes have estimated LLOQs around 1–50 ng/mL, dynamic ranges >100-fold, and limits of quantification varying by serotype. The highest LLOQ of 338.0 ng/mL was observed for serotype 23F. Given that the polysaccharide concentration of each serotype is 50 μg/mL for a current on-market PPV (PNEUMOVAX^®^ 23 product insert), and high concentrations are anticipated in purified monovalent PS encountered during manufacturing of PPVs and PCVs; this sensitivity is certainly sufficient for antigen tracking and quantification in in-process and final vaccine samples. The ULOQ ranged from ~0.9 to 6.5 µg/mL, with the lowest ULOQ of 0.9 μg/mL observed for serotype 12F and the highest ULOQ of 6.5 µg/mL observed for serotype 33F.

Estimated LLOQ, ULOQ, and dynamic ranges for contrived 13-valent Pfizer conjugates and 14-valent EuBiologics conjugates labeled with both the 23-Mix PS and CRM197 labeling strategies are shown in Appendix A. Estimated LLOQs ranged from 1–200 ng/mL, and ULOQ ranged from sub 1 μg/mL to near 3 μg/mL. As observed for the native PS, limits of quantification varied by serotype. In addition, limits of quantification and working ranges were similar for both the PS and carrier protein labeling strategies for a given serotype. Due to the lack of reactivity to Pfizer S18C material with the 23-Mix label, as previously discussed, LLOQ and ULOQ for this material could not be estimated. Of note, early development efforts utilizing longer antigen incubation times of 30 min with active mixing on the ArrayMax at 700 rpm resulted in exquisite sensitivity, with some serotypes approaching the low pg/mL range (data not shown). Because this would have extended the overall time to result for the assay and required several orders of magnitude of up-front dilution to ensure vaccines samples in the working range, the antigen incubation time was significantly reduced to achieve a more relevant working range (most conjugate PCVs are present at 4.4 µg/mL). Therefore, extending the antigen incubation time may enable applications for which better sensitivity is required.

Overall, the VaxArray pneumococcal assay demonstrated vaccine-relevant sensitivity and good dynamic range with both native PS and conjugates and for two detection labeling strategies. These data indicate that the assay is suitable for quantifying serotype-specific polysaccharide and could therefore be applied to a variety of applications throughout PPV and PCV manufacturing, including for monovalent native PS, monovalent conjugate, multivalent bulk, and multivalent final vaccine.

Quantification of free PS over time and storage conditions is another important quantitative application for PCVs. Conjugation of the PS to the carrier protein is critical to immunogenicity, and therefore, tracking stability requires assessing any change in conjugate concentration (typically indirectly measured via an increase in free PS due to degradation of the conjugate) over time. With an appropriate separation method of the conjugate from free PS (manufacturer-specific column-based separation and protein precipitation methods in which the remaining free PS is measured are common), the VaxArray pneumococcal assay could be utilized to measure free PS. Alternatively, because the microarray capture antibodies bind to the capsular PS portion of a conjugate, direct quantification of only material that is conjugated to the CRM197 carrier protein *without* the need for an up-front separation of free and bound (conjugated) PS is enabled through use of the CRM197 labeling strategy.

### 3.4. Precision and Accuracy

Native Polysaccharides. An analysis of user to user and day-to-day accuracy and precision for a 23-valent native PS mixture (ATCC, deposited by Pfizer) was conducted. Three users performed the assay on eight replicates alongside a standard curve of the same material on each of 3 separate days; the data are summarized in Table 3. Data are separated by serotype and user as well as combined for all three users. Accuracy values, expressed as %recovery based on measured concentrations provided by supplier, ranged from 77 to 121%, with an average of 102 (±10)%. Precision of the back-calculated concentration, expressed as the %RSD of the eight replicates, ranged from 6% to 21%, with an overall average precision of 11 (±3)%.

Measurement accuracy and precision was also evaluated on a drug product, specifically PNEUMOVAX^®^ 23. The vaccine was tested by a single user in three separate assay setups. As shown in Table 4, accuracy values ranged from 84% to 112%, with an average of 98 (±8)%. Precision ranged from 7% to 21%, with an average of 12 (±4)%. These data indicate good accuracy and precision for in-process native PS materials as well as final vaccines containing native (unconjugated) PS.

Conjugated Polysaccharide. Initial assessment of precision and accuracy of unadjuvanted conjugate materials indicated similar accuracy and precision to the native PS data shown in Table 3 and Table 4 (data not shown). However, during initial analyses of adjuvanted Pfizer-provided 13-valent mock PCV drug product using the VaxArray pneumococcal assay, significantly lower signals were observed relative to the drug substance without adjuvant. Because of this interference from the aluminum phosphate adjuvant on assay signal, prior to executing additional accuracy and precision studies on adjuvanted conjugates (including final vaccines), a sodium citrate desorption procedure was optimized using 13-valent mock PCV drug products (provided by Pfizer both with and without adjuvant) to investigate whether acceptable %recovery of the conjugates could be achieved post desorption. Figure 4a highlights the VaxArray response in terms of % signal compared to the unadjuvanted drug product as a control generated for the 13-valent adjuvanted drug product both without citrate desorption (black bars) and after citrate desorption (grey bars) using the 23-Mix label. As mentioned above, significant inhibition of signal is observed without desorption. However, signals for all 13 serotypes post desorption were recovered to between 80–120% of the expected signal. Similar results are observed for the anti-CRM197 label as shown in Figure 4b, with 12 of the 13 serotypes (all except S19F) recovering the signal to between 80–120% of expected.

To assess accuracy and precision for additional adjuvanted conjugate materials, EuBiologics-provided 15-valent drug product was compared before and after citrate desorption in a single experimental setup using the 23-Mix label (see Appendix A). Post desorption, every serotype in the EuBiologics drug product represented on the microarray showed acceptable accuracy, ranging from 83% to 118% of expected concentration, and precision for all serotypes showed <20% RSD for replicate measurements. Individual microarray spot morphology was also significantly improved post desorption (see Appendix A).

To evaluate assay accuracy and precision on final conjugate vaccine, PREVNAR 20^TM^ was assessed post citrate desorption with both the 23-Mix Primary Detection Label and anti-CRM197 Primary Detection Label by a single user over three independent assay setups. As shown in Table 5, accuracy ranged from 87% to 105% using the 23-Mix label, with an average of 97 (±6)%. Precision ranged from 3% to 19%, with an average of 10 (±4)%. When using the CRM197 labeling strategy, accuracy ranged from 83% to 102% (average of 93 (±6)%), and precision ranged from 3% to 23% (average of 10 (±4)%). Due to the previously described S23F capture antibody cross-reactivity with Pfizer-provided S19A conjugate, S23F conjugate measurements may be subject to interference by S19A.

Collectively, these data indicate acceptable accuracy and precision for unadjuvanted conjugate materials using both available labeling strategies and that good accuracy and precision are obtained for drug product formulations provided the described citrate desorption protocol is performed prior to VaxArray analysis.

### 3.5. Comparison to Anthrone Assay

Currently, non-specific absorbance-based methods, such as the anthrone and uronic acid assays, are used to quantify total polysaccharide content. To investigate the correlation in assay response between VaxArray and the traditional anthrone assay, EuBiologics native PS and conjugates for serotypes 6A, 18C, and 23F were assessed by both methods. Excellent linear correlation (R values > 0.96) was observed between the signal responses for all six samples evaluated, as shown in Figure 5. Figure 5a shows a representative signal correlation plot for S6A native PS, and Figure 5b shows comparative precision metrics and correlation coefficients for all samples investigated (n = 3 for each sample for both methods). VaxArray precision resulted in average %RSD < 20% for replicate measurements. In contrast, the anthrone assay precision was poorer, with %RSDs as high as 98% for replicate measurements.

Given the strong correlation in signal responses, the VaxArray pneumococcal assay is an attractive alternative to the anthrone assay for total polysaccharide quantification. As noted in the materials and methods section, the anthrone assay measurements required samples to be at higher concentrations, with an additional 100× or 200× dilution needed to ensure measurements were in the VaxArray working range. The VaxArray assay has numerous benefits over the anthrone assay, including significantly improved sensitivity and precision and no hazardous waste production. Furthermore, the anthrone assay and other chemical assays lack any specificity and can only quantify total polysaccharides in the sample, limiting utility to singleplex samples.

## 4. Conclusions

Pneumococcal conjugate vaccines are one of the most complicated biologics to manufacture, requiring hundreds of manufacturing and quality control steps and taking years to produce. The availability of new analytical tools that can significantly simplify multiple quality measurements, such as identity, quantity, and stability of these critical, highly multiplexed vaccines, will enable significant improvements in manufacturing efficiency. The current methodologies utilized for identity and quantification are typically singleplex, requiring a separate test for every serotype present in the vaccine. In addition, most assays currently utilized are internally developed since standardized off-the-shelf kits have not been available.

The VaxArray pneumococcal assay described herein enables simultaneous, multiplexed assessment of 23 serotypes common in PCVs both on the market and in development in less than 30 min.

The assay can detect native PS and CRM197 conjugates with high reactivity and specificity in both multivalent and monovalent samples, enabling use of the assay as an identity test. With an overall average accuracy of 100(±9)% and precision of 11(±4)% using the 23-Mix primary detection label and an overall average accuracy of 97(±6)% and precision of 10(±4)% using the anti-CRM197 primary detection label, the test is quantitative with good accuracy and precision for the measurement of PS content in both native PS and conjugates for both monovalent and multivalent samples, including drug products of PPVs and PCVs, enabling utility during manufacturing and for stability in indicating measurements of free PS. VaxArray is also capable of direct conjugate quantification for CRM197-containing PCVs through use of an anti-CRM197 label.

Though serotype 11A is not currently included due to the lack of a high quality, serotype-specific antibody that performed adequately, we plan to include a S11A capture antibody in a next-generation VaxArray pneumococcal assay. Moreover, while we did compare our assay to the typical anthrone chemical assay in terms of quantification accuracy and precision on several native PS and conjugates samples and showing good signal correlation and improved precision, we were unable to compare our assay to the standard rate nephelometric methods in this specific work but believe this comparison could add value in a future investigation.

Given the number of PCVs currently in development worldwide, we hope the availability of this new, highly multiplexed tool with a rapid time to result will contribute to increased efficiencies in characterization and release testing for these life-saving vaccines.

## Figures and Tables

**Figure 1 vaccines-10-01964-f001:**
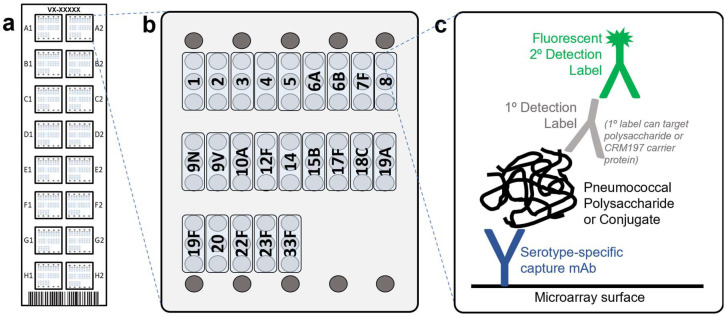
Design of VaxArray pneumococcal assay. (**a**) Schematic representation of the VaxArray pneumococcal assay kit microarray slide showing 16 replicate microarrays, A1-H2 represents the location of each microarray; (**b**) individual microarray layout showing 3 replicate spots for each serotype, with fiducial markers in grey at top and bottom; and (**c**) general assay detection scheme showing serotype-specific capture with primary detection and fluorescent secondary detection label. The primary label can either target polysaccharide or CRM197 carrier protein. (For interpretation of the references to color in this figure legend, the reader is referred to the web version of this article).

**Figure 2 vaccines-10-01964-f002:**
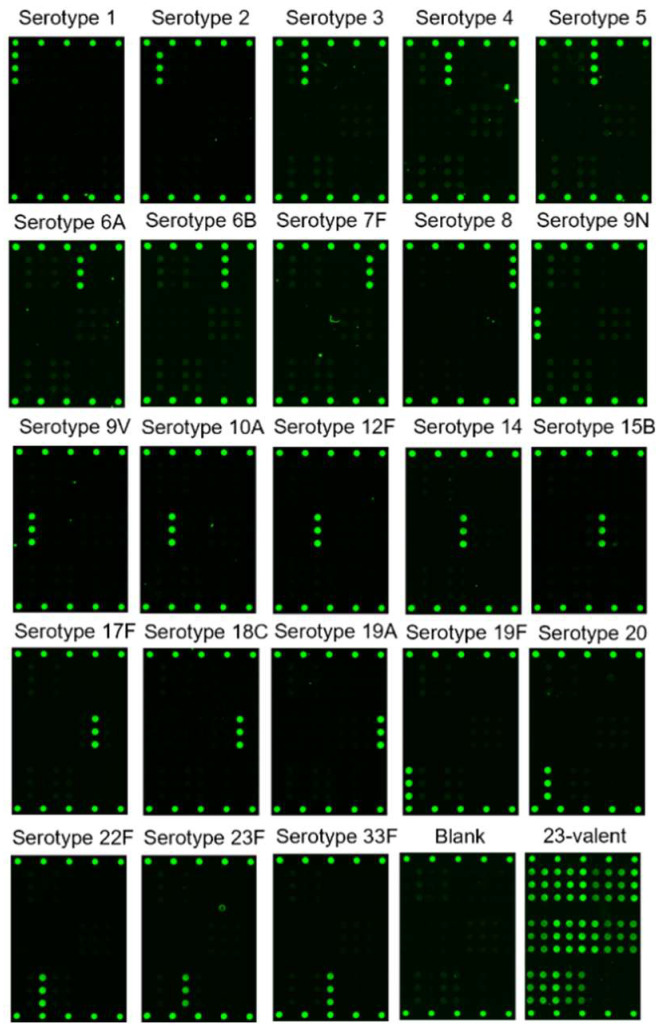
Representative fluorescence microarray images demonstrating reactivity and specificity of the VaxArray pneumococcal assay after the incubation of 2 µg/mL monovalent Pfizer native PS serotypes 1, 2, 3, 4, 5, 6A, 6B, 7F, 8, 9N, 9V, 10A, 12F, 14, 15B, 17F, 18C, 19A, 19F, 20, 22F, 23F, and 33F (ATCC, deposited by Pfizer); a no antigen blank; and a 23-valent mixture containing all serotypes mentioned above.

**Figure 3 vaccines-10-01964-f003:**
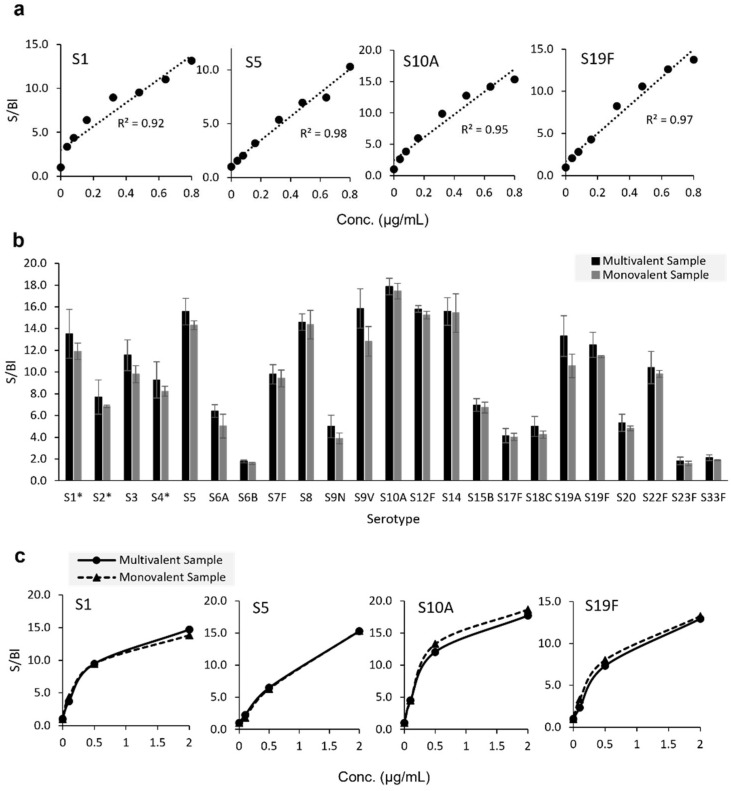
Signal response curves and comparison of response curves for monovalent vs. multivalent analyses. (**a**) Eight-point response curves generated for monovalent native S1, S5, S10A, and S19F labeled using 23-Mix Primary Detection Label. R2 is based on single linear fit (dotted line). (**b**) Signal-to-blank (S/Bl) ratios generated using native PS (ATCC, deposited by Pfizer) at 0.8 μg/mL in contrived 23-valent native polysaccharide mixture and in monovalent form detected by 23-Mix Primary Detection Label. Error bars indicate one standard deviation (n = 3), *: S1, S2, and S4 data are from 400 ms exposure due to signal saturation at 700 ms. (**c**) Response curves were generated using native S1, S5, S10A, and S19F, analyzed monovalently (dotted) and in contrived 23-valent mixture (solid) at 0, 0.1, 0.5, and 2 μg/mL and labeled using 23-Mix Primary Detection Label.

**Figure 4 vaccines-10-01964-f004:**
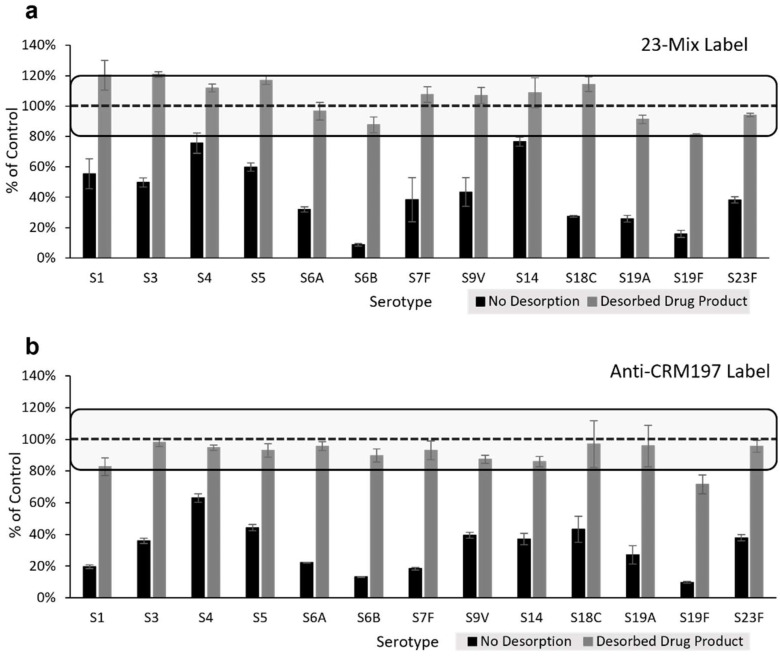
Pfizer 13-valent mock drug product signal before and after citrate desorption detected by 23-Mix Primary Detection Label (**a**) and anti-CRM197 Primary Detection Label (**b**). Results are from three independent citrate desorption experiments, and signals before and after desorption were normalized to the signal of un-adjuvanted drug substance analyzed at the same concentration. Error bars represent ± 1 standard deviation. Dashed lines indicate 100% of control, with the square boxes highlighting %recovery of 80–120%.

**Figure 5 vaccines-10-01964-f005:**
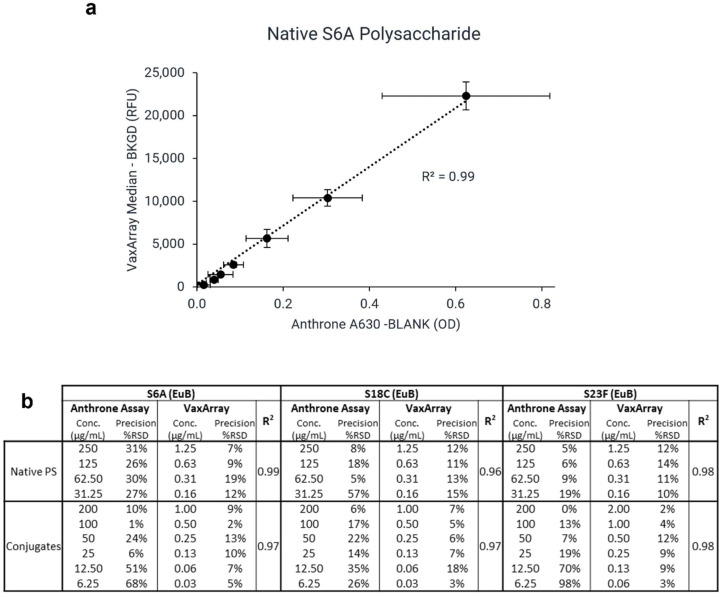
Signal correlation between anthrone and VaxArray assays. (**a**) Signal response correlation for S6A native PS from EuBiologics (EuB). The same serial dilution evaluated by the anthrone assay was diluted 100- to 200-fold to perform VaxArray assay due to VaxArray assay sensitivity. Error bars indicate ± 1 standard deviation (n = 3). (**b**) Precision (%RSD) for anthrone and VaxArray assays and associated R2 value for signal correlation of each sample (n = 3 for each concentration of each method).

**Table 1 vaccines-10-01964-t001:** Native PS reactivity. Signal-to-blank ratios (S/Bl) on the VaxArray 23-valent pneumococcal assay for Pfizer native PS materials (ATCC, deposited by Pfizer) at 2 µg/mL; columns represent each serotype-specific capture antibody, and rows represent each serotype of native PS tested. Text in bold green shows S/Bl ≥ 3 generated for the matched PS and capture antibody. Text in bold black shows S/Bl ≥ 3 generated on off-target capture antibody indicating cross-reactivity.

		Serotype-Specific Microarray Capture Antibody																						
		S1	S2	S3	S4	S5	S6A	S6B	S7F	S8	S9N	S9V	S10A	S12F	S14	S15B	S17F	S18C	S19A	S19F	S20	S22F	S23F	S33F
**Native Polysaccharide Serotype**	**S1**	** 13.8 **	1.0	1.0	1.0	1.0	1.0	1.0	1.0	1.0	1.1	1.1	1.0	1.0	1.0	1.0	1.0	1.0	1.0	1.0	1.1	1.1	1.1	1.1
	**S2**	1.2	** 6.8 **	1.3	1.1	1.0	1.0	1.0	1.1	1.1	1.1	1.0	1.0	1.0	1.0	1.0	1.0	1.1	1.1	1.0	1.0	1.1	1.0	1.0
	**S3**	1.0	1.0	** 7.1 **	1.0	0.9	0.9	0.9	0.9	0.9	1.0	1.0	1.0	1.0	1.0	0.9	1.0	1.0	0.9	0.9	1.1	1.0	1.1	1.2
	**S4**	1.1	1.2	1.1	** 9.3 **	1.0	0.9	0.9	1.0	1.0	1.0	1.0	1.0	1.0	1.0	1.2	1.2	1.1	0.9	1.0	1.3	1.1	1.3	1.3
	**S5**	1.0	1.2	1.1	1.2	** 15.3 **	1.0	1.0	1.0	1.2	1.1	1.1	1.0	1.1	1.1	1.2	1.1	1.0	1.0	1.0	1.4	1.1	1.1	1.2
	**S6A**	1.0	1.1	1.0	1.1	0.9	** 7.0 **	0.9	0.9	0.9	0.9	0.9	0.9	1.0	0.9	1.0	1.1	1.0	0.9	0.9	1.2	1.0	1.1	1.2
	**S6B**	1.1	1.0	1.0	1.0	1.1	1.1	** 3.9 **	1.0	1.0	1.0	1.1	1.2	1.1	1.1	1.1	1.1	1.1	1.1	1.0	1.1	1.2	1.2	1.1
	**S7F**	1.0	1.1	0.9	1.1	0.9	0.9	0.9	** 8.2 **	0.9	0.9	1.0	1.0	1.0	1.0	1.0	1.0	1.0	0.9	0.9	1.1	1.0	1.0	1.2
	**S8**	1.1	1.0	1.1	1.0	1.1	1.1	1.1	1.1	** 14.3 **	1.0	1.1	1.1	1.1	1.1	1.0	1.0	1.0	1.1	1.0	1.0	1.0	1.0	0.9
	**S9N**	1.1	1.2	1.0	1.2	1.0	1.0	0.9	0.9	0.9	** 7.0 **	1.0	1.0	1.0	1.0	1.2	1.3	1.2	1.1	0.9	1.1	1.0	1.1	1.1
	**S9V**	1.2	1.3	1.1	1.1	1.1	1.1	1.1	1.1	1.1	1.1	** 18.7 **	1.4	1.1	1.1	1.1	1.1	1.1	1.0	1.0	1.1	1.0	1.1	1.0
	**S10A**	1.1	1.2	1.1	1.2	1.1	1.1	1.1	1.1	1.1	1.1	1.1	** 18.7 **	1.1	1.1	1.2	1.1	1.1	1.0	1.2	1.2	1.1	1.1	1.1
	**S12F**	1.1	1.1	1.1	0.9	1.0	1.1	1.0	1.1	1.1	1.0	1.0	1.1	** 17.4 **	1.1	0.9	0.9	0.9	1.0	1.0	0.9	1.0	0.9	0.9
	**S14**	1.6	1.1	1.1	1.1	1.1	1.1	1.1	1.1	1.0	1.0	1.0	1.0	1.0	** 11.6 **	1.0	1.0	1.0	1.0	1.0	1.0	1.0	0.9	0.9
	**S15B**	1.2	1.0	1.0	1.1	1.1	1.1	1.1	1.2	1.2	1.1	1.1	1.1	1.1	**3.2**	** 7.3 **	1.2	1.2	1.1	1.1	1.0	1.1	1.0	1.2
	**S17F**	1.1	1.1	1.0	1.1	1.0	1.0	1.0	1.0	1.1	1.0	1.0	1.0	1.0	1.1	1.1	** 6.7 **	1.1	1.1	1.0	1.0	1.0	1.0	1.2
	**S18C**	1.1	1.1	1.0	1.0	1.0	1.1	1.1	1.1	1.1	1.1	1.1	1.0	1.1	1.1	1.1	1.0	** 9.8 **	1.1	1.1	1.0	1.0	1.0	1.2
	**S19A**	1.1	1.1	1.1	1.2	1.1	1.1	1.1	1.1	1.1	1.1	1.1	1.1	1.1	1.1	1.2	1.2	1.2	** 16.9 **	1.1	1.1	1.0	1.1	1.3
	**S19F**	1.1	1.1	1.0	1.2	1.0	1.0	1.0	1.0	1.0	1.0	1.0	1.0	1.0	1.0	1.1	1.1	1.1	1.0	** 13.3 **	1.2	1.0	1.1	1.2
	**S20**	1.2	1.2	1.1	1.2	1.1	1.1	1.1	1.1	1.1	1.2	1.1	1.1	1.2	1.2	1.3	1.5	1.4	1.2	1.2	** 8.1 **	1.2	1.3	1.3
	**S22F**	1.1	1.3	1.2	1.2	1.0	1.0	1.0	1.1	1.0	1.1	1.0	1.2	1.1	1.1	1.5	1.5	1.3	1.1	1.1	1.6	** 13.2 **	1.5	1.4
	**S23F**	1.1	1.1	1.0	1.0	1.0	1.0	1.0	1.1	1.1	1.1	1.1	1.1	1.1	1.1	1.3	1.2	1.2	1.0	1.1	1.4	1.1	** 4.7 **	1.2
	**S33F**	1.0	1.0	1.0	1.0	1.0	1.0	0.9	1.0	1.0	0.9	1.0	1.0	1.0	1.0	1.1	1.1	1.0	1.0	0.9	1.1	1.0	1.1	** 3.4 **

**Table 2 vaccines-10-01964-t002:** Analytical sensitivity and working range for 23-valent Pfizer native PS materials (ATCC, deposited by Pfizer) with 23-Mix Primary Detection Label.

Serotype	Approx. LLOQ (ng/mL)	Approx. ULOQ (μg/mL)	Working Range (x-Fold)	Serotype	Approx. LLOQ (ng/mL)	Approx. ULOQ (μg/mL)	Working Range (x-Fold)
1	2.4	1.0	424	12F	4.8	0.9	182
2	4.5	1.4	315	14	2.9	1.2	426
3	12.4	1.6	131	15B	3.1	1.4	452
4	2.2	1.1	486	17F	2.2	3.6	1631
5	8.3	1.3	154	18C	7.4	>2.9 *	>397 *
6A	31.9	1.9	58	19A	18.3	1.2	65
6B	129.6	5.1	40	19F	10.8	1.3	121
7F	8.3	1.8	223	20	7.1	1.8	259
8	4.1	1.0	239	22F	9.1	1.7	180
9N	34.6	4.7	136	23F	338.0	>2.4 *	>7 *
9V	23.0	1.8	77	33F	40.4	6.5	160
10A	4.2	1.0	233				

***** Materials with “>” symbol had a good linear response over the highest concentration tested, indicating the ULOQ may exceed the value estimated.

**Table 3 vaccines-10-01964-t003:** Accuracy and precision for 23-valent Pfizer native PS materials (ATCC, deposited by Pfizer) over multiple users and multiple days.

Serotype	Expected Conc. (μg/mL)	Accuracy (%Recovery)				Precision (%RSD of Measured Conc.)			
		User 1 (n = 24)	User 2 (n = 24)	User 3 (n = 24)	Overall (n = 72)	User 1 (n = 24)	User 2 (n = 24)	User 3 (n = 24)	Overall (n = 72)
**S1**	0.20	119%	98%	113%	110%	8%	8%	11%	9%
**S2**	0.20	110%	100%	101%	104%	12%	9%	12%	11%
**S3**	0.20	107%	93%	109%	103%	9%	8%	15%	11%
**S4**	0.20	100%	89%	98%	96%	9%	7%	14%	10%
**S5**	0.15	94%	86%	105%	95%	9%	10%	10%	10%
**S6A**	0.25	115%	96%	121%	111%	13%	17%	13%	15%
**S6B**	0.75	89%	77%	100%	89%	11%	13%	16%	13%
**S7F**	0.25	99%	98%	103%	100%	9%	11%	10%	10%
**S8**	0.20	97%	95%	98%	97%	8%	12%	8%	9%
**S9N**	0.63	112%	92%	88%	97%	11%	13%	18%	14%
**S9V**	0.20	113%	99%	93%	101%	9%	12%	15%	12%
**S10A**	0.15	115%	99%	105%	106%	10%	6%	13%	9%
**S12F**	0.20	110%	93%	102%	102%	11%	8%	10%	10%
**S14**	0.15	115%	92%	109%	105%	12%	14%	16%	14%
**S15B**	0.15	110%	97%	106%	104%	9%	8%	10%	9%
**S17F**	0.25	114%	93%	111%	106%	15%	9%	7%	10%
**S18C**	0.20	109%	93%	113%	105%	11%	10%	12%	11%
**S19A**	0.20	100%	88%	98%	95%	8%	9%	8%	8%
**S19F**	0.20	118%	108%	97%	108%	11%	18%	20%	16%
**S20**	0.15	117%	99%	99%	105%	8%	11%	16%	12%
**S22F**	0.25	110%	92%	95%	99%	8%	11%	16%	11%
**S23F**	0.38	115%	80%	108%	101%	15%	18%	21%	18%
**S33F**	1.00	108%	89%	100%	99%	10%	8%	12%	10%
**Overall**		102 ± 10%				11 ± 3%			

**Table 4 vaccines-10-01964-t004:** Accuracy and precision for PNEUMOVAX^®^ 23 using 23-Mix Primary Detection Label. Expected concentration of each serotype is 0.2 μg/mL.

Serotype	Accuracy (%Recovery)	Precision (%RSD)	Serotype	Accuracy (%Recovery)	Precision (%RSD)
**S1**	112%	13%	S12F	104%	21%
**S2**	104%	10%	S14	110%	14%
**S3**	100%	11%	S15B	95%	7%
**S4**	110%	12%	S17F	90%	11%
**S5**	101%	9%	S18C	91%	7%
**S6B**	84%	18%	S19A	87%	8%
**S7F**	95%	8%	S19F	99%	17%
**S8**	105%	12%	S20	95%	12%
**S9N**	92%	13%	S22F	100%	10%
**S9V**	97%	15%	S23F	N/A *	N/A *
**S10A**	108%	8%	S33F	87%	19%
**Overall**	**Accuracy** 98 ± 8%		**Precision** 12 ± 4%		

* Not evaluated, as the concentration present is below its estimated LLOQ.

**Table 5 vaccines-10-01964-t005:** Accuracy and precision for desorbed Prevnar 20^TM^ using both 23-Mix and anti-CRM197 Primary Detection Labels. Expected concentration of each serotype is 0.2 μg/mL (except S6B, which is present at 0.4 μg/mL).

Serotype	23-Mix Label		Anti-CRM197 Label	
	Accuracy (%Recovery)	Precision (%RSD)	Accuracy (%Recovery)	Precision (%RSD)
**S1**	94%	15%	102%	9%
**S3**	87%	3%	101%	12%
**S4**	97%	9%	102%	8%
**S5**	105%	6%	96%	7%
**S6A**	99%	16%	90%	8%
**S6B**	95%	17%	91%	23%
**S7F**	93%	19%	89%	10%
**S8**	101%	8%	93%	12%
**S9V**	101%	8%	98%	3%
**S10A**	104%	9%	95%	13%
**S12F**	105%	8%	90%	8%
**S14**	96%	13%	91%	13%
**S15B**	89%	9%	86%	9%
**S18C**	103%	6%	96%	8%
**S19A**	96%	4%	92%	11%
**S19F**	95%	15%	84%	14%
**S22F**	100%	7%	90%	8%
**S23F**	90%	13%	92%	11%
**S33F**	89%	13%	83%	12%
**Overall**	97 ± 6%	10 ± 4%	93 ± 6%	10 ± 4%

## Data Availability

All relevant data from this study are available from the authors.

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
