# Peer review of "30-Minute Highly Multiplexed VaxArray Immunoassay for Pneumococcal Vaccine Antigen Characterization"

_vaccines, 2022, doi:10.3390/vaccines10111964_

Round 1

Reviewer 1 Report

Thiis an exciting, well planned and researched and meticulously presented study.

I only have some minor points to further underline the contributions the authors are making.

Introduction

Concise and sets out the problem.

Methods

“While serotype 220 11A was targeted for inclusion, poor antibody performance and lack of high-quality alter- 221 natives available at the time of development precluded its inclusion.”

[This is a widely recognised problem. What are the quantitative implications for VE?]

“In addition, a polysaccharide label mixture to enable simultaneous 227 23-valent PS detection was carefully optimized (23-Mix Primary Detection Label, VXPN- 228 7671).’

[Please describe in detail what “carefully optimized” means. This will help fellow researchers]

Results

“To assess response similarity over a broader con- 268 centration range, all 23 serotypes were tested at 3 different concentrations. Similar re- 269 sponses were observed at all tested concentrations.”

[What are the implications for the amount of each serotype provided per vaccinee?]

Conclusions

“The assay can detect native PS and CRM197 conjugates with high 473 reactivity and specificity in both multivalent and monovalent samples, enabling use of the 474 assay as an identity test. The test is quantitative with good accuracy and precision for 475 the measurement of PS content in both native PS and conjugates for both monovalent and 476 multivalent samples, including drug products of PPVs and PCVs, enabling utility during 477 manufacturing and for stability indicating measurements of free PS. VaxArray is also ca- 478 pable of direct conjugate quantification for CRM197-containing PCVs through use of an 479 anti-CRM197 label”

[Could you please provide numerical data for this statement “good accuracy and precision” so that readers will carry away your key message].

Author Response

Point by Point Response to Reviewer #1 Feedback

1. This is a widely recognized problem. What are the quantitative implications for VE? (“While serotype 11A was targeted for inclusion, poor antibody performance and lack of high-quality alternatives available at the time of development precluded its inclusion.”)

InDevR Response: Thank you for your comment, but we are not sure how the lack of an available monoclonal capture antibody for serotype 11A is related to vaccine efficacy, as the polyclonal antibody response generated in vivo post-vaccination (which does not have to exhibit antibody specificity to a specific serotype) is quite different than the requirements of a monoclonal antibody to function well in our assay. Existing data for on-market pneumococcal vaccines such as Prevnar20 clearly show that appropriate antibody responses against included serotypes are generated post-vaccination via assays such as ELISA and opsonophagocytosis (OPA)-based assays. We are again making a clear distinction between the polyclonal antibody response generated in vivo post-vaccination, and the generation/availability of a sufficient monoclonal, highly purified, commercial antibody that binds serotype 11A capsular polysaccharide. The inavailability of an appropriate monoclonal could be due to numerous reasons including inability to isolate and purify a single clone that meets our assay requirements of being absolutely specific to serotype 11A.

2. Please describe in detail what “carefully optimized” means. This will help fellow researchers. (“In addition, a polysaccharide label mixture to enable simultaneous 23-valent PS detection was carefully optimized (23-Mix Primary Detection Label, VXPN- 228 7671)”)

InDevR Response: Thank you very much for this suggestion. We have added more details about the optimization of the polysaccharide label mixture in the Materials and Methods section. Please see page 4, lines 151 through 160 in the redlined manuscript. This change required additional minor edits on page 6, lines 241 through 246 in the initial location of the description of the label concentration optimization.

3. What are the implications for the amount of each serotype provided per vaccine? (“To assess response similarity over a broader concentration range, all 23 serotypes were tested at 3 different concentrations. Similar responses were observed at all tested concentrations.”)

InDevR Response: The intent of the data in Figure 1c on page 5 that the reviewer references was to show that the response curves for a specific serotype are essentially independent of whether the sample is run monovalently or multivalently (and therefore that no interference is present when analyzing a multivalent sample). The 3 concentrations tested in these limited 4-point response curves (3 concentrations and a blank) were meant to bracket the limits of quantification of the assay. Given that the concentration of each serotype in current on-market conjugate vaccines is around 4 μg/mL (with a few exceptions) and current on-market polysaccharide vaccines is 50 μg/mL, a simple dilution of either type of vaccine sample moves that analysis into the appropriate working range, and the sensitivity is adequate. This question of the dynamic range being vaccine-relevant is presented in more detail in the Limits of Quantification and Dynamic Range section beginning on page 9 line 331. Specifically, we discuss polysaccharide vaccines in lines 336-340, and have added a phrase more clearly calling out the conjugate amount in PCVs on page 10, lines 359-360.

4. Could you please provide numerical data for this statement “good accuracy and precision” so that readers will carry away your key message. (“The assay can detect native PS and CRM197 conjugates with high reactivity and specificity in both multivalent and monovalent samples, enabling use of the assay as an identity test. The test is quantitative with good accuracy and precision for the measurement of PS content in both native PS and conjugates for both monovalent and multivalent samples, including drug products of PPVs and PCVs, enabling utility during manufacturing and for stability indicating measurements of free PS. VaxArray is also capable of direct conjugate quantification for CRM197-containing PCVs through use of an anti-CRM197 label”)

InDevR Response: We appreciate this suggestion and agree that adding specifics for this statement in the Discussion section will help readers put the VaxArray Assay performance in context. Therefore, we have calculated the average accuracy and precision over all targeted serotypes presented in the paper using 23-Mix anti-PS detection label and anti-CRM197 detection label and have added this information in the relevant area of the Discussion section (please see page 16, lines 495-497 of the redlined manuscript).

Reviewer 2 Report

I was invited to revise the paper entitled "30-Minute Highly Multiplexed VaxArray Immunoassay for Pneumococcal Vaccine Antigen Characterization". This study presents the performance metrics of a new 23-valent VaxArray Pneumococcal Assay. The strenght of this methodology was the short time from the sample to the final results (30-min), with high reactivity and serotype-specificity to polysaccarides. In addition, strong correlation was find with other assay was reported.

These results are relevant and could have a strong impact.

I have only some minor observations:

- Authors should add a "Statistical Analysis" section in Methods;

- Authors should divide results section from discussions. In addition, Authors should compare their results with similar tools and They should add a strenght and limitation section.

Author Response

Point by Point Response to Reviewer #2 Feedback

1. Authors should add a "Statistical Analysis" section in Methods

InDevR Response: Thank you very much for this comment. We addressed this by adding a short “Statistical analysis” section in the Materials and Methods section on page 6, lines 224-229 of the redlined manuscript.

2. Authors should divide results section from discussions.

InDevR Response: We appreciate this comment but are unable to separate these two sections given timeline limitations on the revision. Because we have co-authors from Pfizer, a significant change in the text would require a more extensive internal review by Pfizer prior to re-submission of the manuscript. We notified the editorial office of this delay, but there was a request for us to still re-submit the revised manuscript within the 5-day turnaround time if at all possible, and therefore, we have elected not to institute this change. Given the content would not substantially change (just the location of the existing content), we hope this is an acceptable response to the reviewer and the editorial office.

3. Authors should compare their results with similar tools.

InDevR Response: This is a fantastic suggestion for future work but is outside the scope for inclusion in the current manuscript. We did compare our assay to the typical anthrone chemical assay for polysaccharide quantification and showed that our assay signal is highly correlated with anthrone assay signal but offers improved precision. Another assay that would be interesting in terms of comparison is rate nephelometry, another commonly utilized polysaccharide quantification methodology—however, we do not have access to this technology and were unsuccessful at finding a willing collaborator to perform a side-by-side comparison for this particular study. We will consider such an inclusion in our future work and thank you for your insights.

4. They should add a strength and limitation section

InDevR Response: This comment is very helpful to improve manuscript impact. We have added a paragraph in the Conclusions section page 16, lines 503 through 510 to highlight the limitations. The assay strengths are highlighted on page 16, lines 490 through 502 of the Conclusions section.